Geoscientific
Instrumentation
Methods and
Data Systems

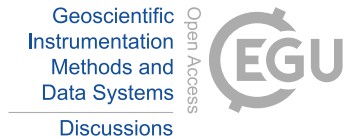

Discussions

# A 7-year dataset for driving and evaluating snow models at an arctic site (Sodankylä, Finland)

R. Essery[1], A. Kontu[2], J. Lemmetyinen[3], M. Dumont[4], and C. B. Ménard[3]

[1]School of GeoSciences, University of Edinburgh, Edinburgh EH9 3FE, UK
[2]Arctic Research Unit, Finnish Meteorological Institute, 99600 Sodankylä, Finland
[3]Finnish Meteorological Institute, 00101 Helsinki, Finland
[4]Météo-France-CNRS, CNRM-GAME UMR 3589, CEN, Grenoble 38000, France

Received: 7 December 2015 – Accepted: 15 December 2015 – Published: 19 January 2016

Correspondence to: R. Essery (richard.essery@ed.ac.uk)

Published by Copernicus Publications on behalf of the European Geosciences Union.

## GID

doi:10.5194/gi-2015-35

**A 7-year dataset for driving and evaluating snow models**

R. Essery et al.

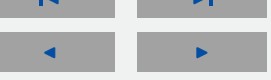



## Abstract

Datasets derived from measurements at Sodankylä in the Finnish Arctic that can be used for driving and evaluating snow models are presented. The driving datasets comprise all of the meteorological variables required as inputs for physically-based snow models at hourly intervals: incoming solar and longwave radiation, snowfall and rainfall rates, air temperature, humidity, wind speed and atmospheric pressure. Two versions of the driving data are provided: one using radiation and wind speed measurements made above the height of the trees around the clearing where the evaluation data were measured and one with adjustments for the influence of the trees on conditions close to the ground. The available evaluation data include automatic and manual measurements of bulk snow depth and snow water equivalent, and profiles of snow temperature, snow density and soil temperature. A physically-based snow model is driven and evaluated with the datasets to illustrate their utility. Shading by trees extends the duration of snow cover on the ground by several days a year.

## 1   Introduction

Many studies have used meteorological data to drive snow models and meteorological or hydrological data to evaluate model performance at instrumented sites. These studies have often only used limited periods of driving data (e.g. two winters for several sites in Essery et al., 2009) or limited evaluation data (e.g. infrequent manual measurements of snow mass in Slater et al., 2001). Recently, valuable datasets have been published with multiple years of driving data and multiple sources of evaluation data for several snow research sites: Reynolds Mountain East in the Owyhee Mountains of Idaho (Reba et al., 2011), Col de Porte in the Chartreuse Mountains of France (Morin et al., 2012), the Senator Beck Basin in the San Juan Mountains of Colorado (Landry et al., 2014) and Weissflujoch in the Plessur Alps of Switzerland (Schmucki et al., 2014).

# GID

doi:10.5194/gi-2015-35

**A 7-year dataset for driving and evaluating snow models**

R. Essery et al.

All of these are high-elevation, mid-latitude sites; there has been a lack of comparable datasets that could be used for evaluating snow models at high latitudes.

Snow models operating on energy balance principles form components of land surface models that are used to provide energy and moisture flux boundary conditions for the atmosphere in numerical weather prediction and climate models, but they can also be driven with measured meteorological data. The typical input data required are downwelling shortwave and longwave radiation fluxes, precipitation rate, air temperature, humidity, wind speed and atmospheric pressure. All of these variables can be measured with low-power instruments, but all are challenging to measure in cold and snowy environments where instruments can be covered by snow or ice and access for maintenance may be difficult. Model driving data have to be continuous, so gap filling is required if instrument or power failures occur. Data timesteps have to be somewhat shorter than a day (often 30 min or 1 h) if situations in which snow melts during the day and refreezes at night are to be explicitly represented.

This paper presents model driving and evaluation datasets collated from measurements made at the Finnish Meteorological Institute's Arctic Research Centre (FMI-ARC) over the 7-year period starting on 1 October 2007. Descriptions are given of the site, the instrumentation, gap filling used to construct a continuous driving dataset and adjustments of above-canopy measurements to allow for influences of shading by trees on below-canopy conditions. Comparisons of model simulations with evaluation data are presented as an illustration of data use and as a quality-control check on the data.

## 2 Site

FMI-ARC (67.368° N, 26.633° E, 179 m a.s.l.) is collocated with the Sodankylä Geophysical Observatory beside the Kitinen river, about 6 km south of the town of Sodankylä in northern Finland. Snow typically lies from October until May, reaching maximum depths between 60 and 100 cm each winter. Air temperatures can fall below

−30 °C in winter, but the sun only remains entirely below the horizon for a few days in December. Meteorological measurements have been made at this site since 1908. Current instrumentation includes an automatic weather station and an upper air sounding station (World Meteorological Organization index number 02836) which transmit data on the Global Telecommunication System for use by numerical weather prediction centres. In addition to regular measurement programmes, the Sodankylä area has been used in many remote sensing missions and field campaigns, including the Nordic Snow Radar Experiment (NoSREx), the Snow Reflectance Transition Experiment (SnoRTEx) and the Solid Precipitation Intercomparison Experiment (SPICE).

Figure 1a is an aerial orthophotograph of the site. The area around FMI-ARC is level and forested, predominantly with pine trees around 15 m tall, but many measurements are made in clearings or in a large wetland area to the east of the site. Driving data for this paper are taken from the automatic weather station (Fig. 1b) and a nearby radiometer tower (Fig. 1c) with instruments that are calibrated annually. Evaluation data are taken from an Intensive Observation Area (Fig. 1d) that was established for NoSREx. A list of many other observations not discussed in this paper can be found at http://litdb.fmi.fi/index.php.

## 3  Driving data and gap filling

All of the meteorological variables necessary for model driving are measured by the AWS and the radiometer tower at FMI-ARC with the instruments and at the heights listed in Table 1; note that radiation and wind measurements are made at heights above the forest canopy. For the 7-year period collated here, less than 1 % of hourly data (visible as red points in Fig. 2) are missing for any variable with the exception of longwave radiation; the longest period of missing data is a 52-day gap in the longwave radiation measurements from 10 September to 31 October 2011 because of a faulty power supply.

**GID**

doi:10.5194/gi-2015-35

**A 7-year dataset for driving and evaluating snow models**

R. Essery et al.

Measurements from the AWS and the radiometer tower are used for driving data whenever they are available, but gaps have to be filled to form a complete driving dataset. Gaps of 4 h or shorter are filled by linear interpolation. For shortwave radiation, air temperature, humidity and wind speed, longer gaps are filled with data from nearby

5 instruments. No alternative longwave radiometer was operating at FMI-ARC for the full period, so longwave gaps are filled using ERA-Interim reanalyses (Dee et al., 2011). Longwave radiation fluxes in ERA-Interim are produced by short-range forecasts that can be expected to be accurate if the analysed vertical profiles of temperature and humidity in the atmosphere are accurate. Data from both the surface synoptic station

10 and the upper air station at Sodankylä are available for assimilation in reanalyses, and ERA-Interim compares well with the in situ measurements; the longwave radiation measurements and forecasts have a root mean square difference of $26.7 \, \mathrm{W\,m^{-2}}$ and a correlation coefficient of 0.88 for periods when both are available (a scatter plot is included as supplementary material). Direct measurements of longwave radiation are

15 rarely available for cold regions and snow models are known to be sensitive to longwave driving data (Raleigh et al., 2015a); having near-continuous longwave measurements is therefore a distinct advantage of the FMI-ARC site.

Seven-year series of gap-filled hourly data are shown in Fig. 2 for all of the driving variables apart from precipitation. Measuring solid precipitation is particularly chal-

20 lenging, and uncertainties in snowfall inputs are a major source of uncertainty in snow model outputs (Raleigh et al., 2015b). It is total precipitation that is usually measured, but this has to be partitioned into snow and rain, either in the driving data for mass balance calculations or by the model. This is usually done by selecting a threshold or function of air temperature or wet-bulb temperature discriminating between rain and

25 snow (Auer, 1974; Sims and Liu, 2015). Figure 3a shows the annual average snowfall partitioned from total precipitation for Sodankylä with varying temperature or wet-bulb temperature thresholds; the snowfall is not very sensitive to the choice of temperature or wet-bulb temperature as a predictor because humidity is usually high during precipitation, but it is sensitive to the choice of threshold because a significant amount of

**GID**

doi:10.5194/gi-2015-35

**A 7-year dataset for driving and evaluating snow models**

R. Essery et al.

precipitation falls at temperatures close to 0 °C. With precipitation classified as snow for temperatures lower than 2 °C, Fig. 3b shows that the cumulated amount of snowfall is less than the maximum observed snow water equivalent (SWE) on the ground in some winters (by 37 % in 2007–2008) and greater in others (by 8 % in 2011–2012). Because the site is cold and little melting of snow occurs in autumn or winter, the cumulated snowfall should be close to the amount of snow on the ground at points that are not effected by canopy interception or wind redistribution. Snowfall data are therefore scaled by the factor required to match the maximum measured SWE each winter; cumulated snowfall then also matches the rate of accumulation on the ground quite well, as shown in Fig. 3b.

The snow measurement points in the IOA (Fig. 2d) are not directly beneath trees, so snow accumulation there will not be greatly affected by canopy interception, but they are shaded from direct solar radiation by nearby trees. The presence of the trees will also increase the incoming longwave radiation and decrease the wind speed near the ground relative to more open locations. Measurements above the forest canopy height do not take these influences into account. To allow the use of snow models without representations of forest canopies, radiation fluxes and wind speed are adjusted in a modified driving dataset. From the hemispherical image of the canopy at the IOA in Fig. 4a, the sky view fraction is estimated as $f_v = 0.8$ and a transmissivity $\tau$ for direct solar radiation is calculated by determining the fraction of each hour for which the sun would be blocked by the canopy. Modified solar radiation is given by

$$SW' = f_v SW_{dif} + \tau(SW - SW_{dif}) \tag{1}$$

where SW and $SW_{dif}$ are the measured incoming global and diffuse solar radiation (Reid et al., 2014). Longwave radiation is modified by assuming that the canopy temperature can be approximated by the air temperature (Essery et al., 2008) so that

$$LW' = f_v LW + (1 - f_v)\sigma T^4 \tag{2}$$

**GID**

doi:10.5194/gi-2015-35

**A 7-year dataset for driving and evaluating snow models**

R. Essery et al.



where LW is the measured incoming longwave radiation, $\sigma = 5.67 \times 10^{-8}\,\mathrm{W\,m^{-2}\,K^{-4}}$ is the Stefan–Boltzmann constant and $T$ is the air temperature in Kelvin. The resulting decreases in solar radiation and increases in longwave radiation are shown in Fig. 4b.

An anemometer installed temporarily at 2 m height above the ground close to the IOA for 7 days in March 2012 recorded an average wind speed that was a fraction 0.35 of the wind speed at 22 m height (equal to the ratio given by a logarithmic wind profile with a roughness length of 0.55 m). This ratio is used to scale the wind speed in the modified driving data set. There is no permanently installed anemometer below the canopy height at the IOA, so the wind adjustment is highly uncertain. Because the wind is rarely strong enough to move snow in the IOA and snowmelt is dominated by radiation in spring, however, it is expected that models will not be highly sensitive to the wind adjustment.

## 4 Evaluation data

Physically-based snow models may include snow temperature, mass, density, liquid water content and grain size in layers as state variables. Predicted fluxes will include reflected shortwave radiation, emitted longwave radiation, sensible and latent heat fluxes to the atmosphere, and conducted heat flux and drainage of water at the base of the snowpack. Snow properties that have to be predicted include albedo and thermal conductivity. Measurements of any state variable, flux or property may be used as evaluation data for models, and the measurements need not be continuous; measured and modelled variables can be compared at whatever times for which measurements are available.

FMI-ARC data that will be used in the model evaluation are listed in Table 2. Again, many more measurements are made in the IOA in addition to those discussed here, including snow grain size, hardness, wetness and microwave brightness temperatures. The microstructure of snow samples taken during special experiments has been measured in great detail by X-ray computed tomography (Proksch et al., 2015). Outgo-

**GID**

doi:10.5194/gi-2015-35

**A 7-year dataset for driving and evaluating snow models**

R. Essery et al.

ing radiative and turbulent flux measurements are made above the canopy height at FMI-ARC, so they would be most useful for evaluating models that include vegetation canopies.

Snow depth and SWE are measured in the IOA both manually about once a week and many times daily with automatic instruments. These measurements are compared in Fig. 5. The output of the experimental SWE sensor, which works by measuring the attenuation of gamma radiation from a source beneath the snow, is noisy but tracks the manual measurements well after calibration and averaging. Snow accumulation varies spatially. Figure 6 compares the snow depth in the IOA for the winter of 2012–2013 with snow depths measured in the forest beside the IOA and 900 m to the northeast on the wetland. The snow was deepest throughout the winter and melted latest in the IOA. Some snow is intercepted by the forest canopy as it falls and can sublimate, reducing the depth of snow on the forest floor. Wind can remove and compact snow in the open wetland area, again reducing the snow depth. Differences in snow accumulation and melt rates lead to differences in the persistence of snow cover at different sites; the measured snow depth fell to zero on 3 May 2013 on the wetland, 6 May in the forest and 13 May in the IOA. Photographs of the IOA in Fig. 7 show small-scale variations in cover as the snow melts. Bare patches first appear around the bases of trees, and the snow lies longest at the shady side of the clearing.

Snow temperatures are measured continuously by an array of thermistors supported on a stick that becomes buried in the snow and intermittently by inserting a stem thermometer into the snow face when pits are dug. Both methods are subject to biases; it was observed that the thermistor stick interferes with the accumulation of snow and can form a depression up to 30 cm deep in the snow surface, and digging a snow pit brings air into contact with snow at the base of the snowpack. Density is measured by weighing snow samples of known volume from snow pits and also by a dielectric method that relates density and wetness to the measured permittivity of snow (Sihvola and Tiuri, 1986). It is observed that the dielectric method generally gives lower densities than gravimetric sampling of snow at Sodankylä.

**GID**

doi:10.5194/gi-2015-35

**A 7-year dataset for driving and evaluating snow models**

R. Essery et al.

## 5  Model results

Preliminary versions of the driving and evaluation datasets were used in a study with the JULES land surface model by Ménard et al. (2015). The above-canopy and modified driving datasets are used here to drive Crocus (Vionnet et al., 2012), which is a detailed multi-layer snowpack model originally developed for avalanche forecasting in the French mountains. Although physically-based, some of the processes in Crocus have been parametrized using experimental results from the mid-latitude site at Col de Porte (45.3° N, 5.8° E, 1325 m a.s.l.), which is much warmer than Sodankylä in winter and has heavier snowfall.

Figure 8 compares Crocus simulations driven by the above-canopy and below-canopy datasets with measurements of snow depth, SWE and soil temperature. Simulated snow depths are generally close to the measurements but are sometimes overestimated after snowfall because of Crocus predicting densities for fresh snow that are lower than observed at Sodankylä. Simulated SWE follows the measurements during the accumulation periods, which is to be expected because of the lack of mid-winter melt and the scaling of the snowfall in the driving data to the SWE measurements. Snowmelt starts at about the right time each spring in the simulations but then proceeds faster than observed. The modified driving data reduces melt rates; simulations with the above-canopy driving data remove the snow on average 13 days earlier than the snow disappearance dates inferred from the ultrasonic depth gauge at the IOA, but simulations with the modified below-canopy driving data remove the snow on average only 6 days earlier than observed. As shown by Fig. 7, the dates of snow disappearance can differ by two weeks even over short distances in reality; this spatial variability is not represented by a one-dimensional model such as Crocus. Simulated soil temperatures have cold spikes that are greater than observed at the start of some winters but then remain close to 0 °C once the snowpack has become established. Measured soil temperatures also show a strong influence of insulation by snow but can fall a couple of degrees lower than the simulations in late winter.

The frequent snow pit measurements in the IOA and the multi-layer outputs of Crocus give a large amount of data for comparison. Profiles of temperature and density for 140 snow pits dug between 7 December 2009 and 14 May 2014 are plotted in a supplement, but the evolution of the snowpack over the winter of 2012–2013 alone is shown in Fig. 9. Snow pits were dug once a week, usually on Tuesdays but sometimes on a Wednesday or Thursday, for the 28 weeks between 31 October 2012 and 7 May 2013. Simulations and measurements both show temperatures remaining close to 0 °C at the base of the snowpack with periods of much colder temperatures in snow layers close to the surface. The snow then rapidly warms and becomes wet and isothermal at 0 °C when melt begins in spring. Density generally increases with depth in the snowpack and with time, again increasing rapidly once the snow becomes wet.

Quantitative comparisons between simulated and measured profiles of snow properties are complicated by differences in simulated and measured snow depths. Simply making scatter plots (Fig. 10) of variables at the same heights above the ground and at the same times shows strong correlations of 0.80 between simulated and measured snow temperatures and 0.74 for densities. The simulated temperatures tend to be higher than observed for the warmer temperatures found near the base of the snowpack.

## 6 Conclusions

Data from the FMI Arctic Research Centre at Sodankylä have been used to construct, for the first time at an arctic site, datasets that will allow driving of snow models for multiple years and evaluation of model outputs against multiple types of observations. There are some gaps in the data, but the availability of additional instruments and high-quality atmospheric reanalyses give confidence in the filling of gaps to provide continuous driving data. The utility of the datasets has been demonstrated by driving the Crocus snow model and evaluating its outputs against snow depth, SWE, snow density, snow temperature and soil temperature measurements. The phys-

**GID**

doi:10.5194/gi-2015-35

**A 7-year dataset for driving and evaluating snow models**

R. Essery et al.

ical basis of the model allows it to perform well in an arctic environment very different to the mid-latitude mountain environments for which it was first developed. It is intended that Sodankylä will be used as a reference site in an upcoming evaluation of snow simulations in Earth System models (http://www.climate-cryosphere.org/activities/targeted/esm-snowmip). The datasets can be downloaded from the FMI litdb archive at http://litdb.fmi.fi/ESMSnowMIP.php.

*Acknowledgements.* The staff at FMI-ARC are thanked for data collection and maintenance of instruments. Collection of evaluation data in the IOA was supported by ESA ESTEC contract 22671/09/NL/JA/ef. Visits to FMI-ARC by the first author were supported by NERC grant NE/H008187/1 and ESA ESTEC contract 23103/09/NL/JC. Samuel Morin and Matthieu Lafaysse assisted with the Crocus simulations. The orthophotograph in Fig. 1a was supplied by the Finnish Geospatial Research Institute.

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

◄          ►│

◄          ►

Back          Close

Full Screen / Esc

Discussion Paper | Discussion Paper | Discussion Paper | Discussion Paper | Discussion Paper

## GID

doi:10.5194/gi-2015-35

**A 7-year dataset for driving and evaluating snow models**

R. Essery et al.

**Table 1.** Instruments and missing data for meteorological driving variables between 1 October 2007 and 30 September 2014.

| Variable | Instrument | Height | Missing data |
|----------|------------|--------|--------------|
| Precipitation | Vaisala VRG101 until September 2013, then OTT Messtechnik Pluvio2 | 1 m | 0.67 % |
| Air pressure | Vaisala PTB201A | 1 m | 0.11 % |
| Air temperature | Pentronic PT100 | 2 m | 0.11 % |
| Relative humidity | Vaisala HMP35D | 2 m | 0.19 % |
| Global solar radiation | Kipp & Zonen CM11 | 14 m | 0.61 % |
| Diffuse solar radiation | Kipp & Zonen CM11 with shading ball | 14 m | 0.55 % |
| Longwave radiation | Kipp & Zonen CG4 | 14 m | 8.65 % |
| Wind speed | Vaisala WAA25 | 22 m | 0.12 % |

# GID

doi:10.5194/gi-2015-35

**A 7-year dataset for driving and evaluating snow models**

R. Essery et al.

Discussion Paper | Discussion Paper | Discussion Paper | Discussion Paper | Discussion Paper |

**Table 2.** Evaluation data from the IOA.

| Variable | Instrument |
| --- | --- |
| Snow depth | Campbell Scientific SR50 |
| | Manual sampling |
| Snow water equivalent | Astrock Gamma Water Instrument |
| | Manual sampling |
| Snow density profiles | Toikka Snow Fork sampling at 10 cm height increments from 10 September 2009 |
| | Manual sampling at 5 cm height increments from 7 December 2009 |
| Snow temperature profiles | Campbell Scientific 107-L at 10 cm height increments from 9 June 2011 |
| | Manual sampling at 10 cm height increments |
| Soil temperature profiles | Decagon Devices 5TE at 5, 10, 20, 40 and 80 cm depths from 9 June 2011 |

# GID

doi:10.5194/gi-2015-35

**A 7-year dataset for driving and evaluating snow models**

R. Essery et al.

Full Screen / Esc

Discussion Paper | Discussion Paper | Discussion Paper | Discussion Paper

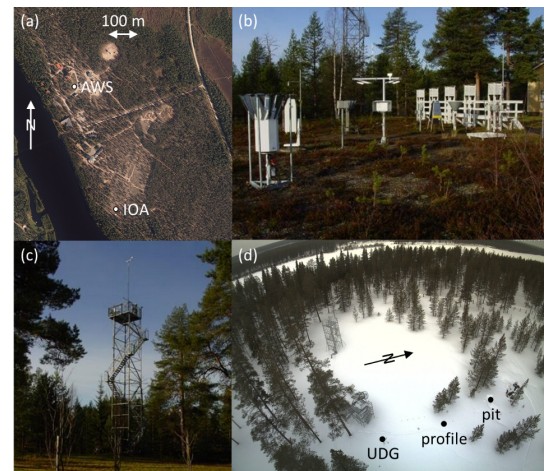

**Figure 1. (a)** Orthophotograph of the FMI-ARC site, showing the locations of the automatic weather station (AWS) and the Intensive Observation Area (IOA). **(b)** The automatic weather station, with the radiometer tower in the background. **(c)** The radiometer tower. **(d)** The IOA, showing the locations of the ultrasonic depth gauge (UDG), the snow temperature profile and the snow pit for manual measurements.

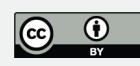

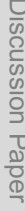

# GID

doi:10.5194/gi-2015-35

**A 7-year dataset for driving and evaluating snow models**

R. Essery et al.

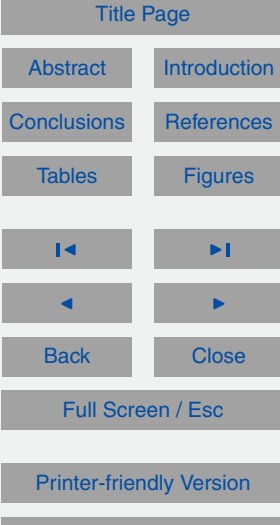

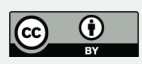

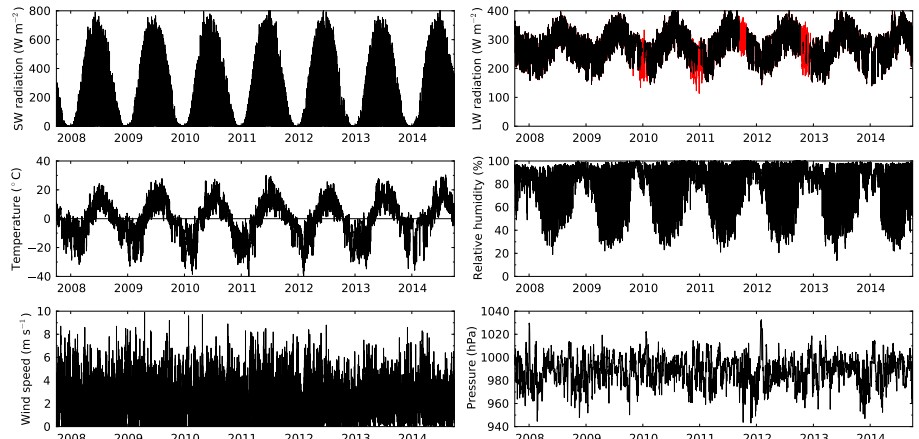

**Figure 2.** Hourly timeseries of shortwave radiation, longwave radiation, air temperature, relative humidity, wind speed and pressure. Longwave radiation data points in red are from reanalyses.

Discussion Paper | Discussion Paper | Discussion Paper | Discussion Paper | Discussion Paper |

# GID

doi:10.5194/gi-2015-35

**A 7-year dataset for driving and evaluating snow models**

R. Essery et al.



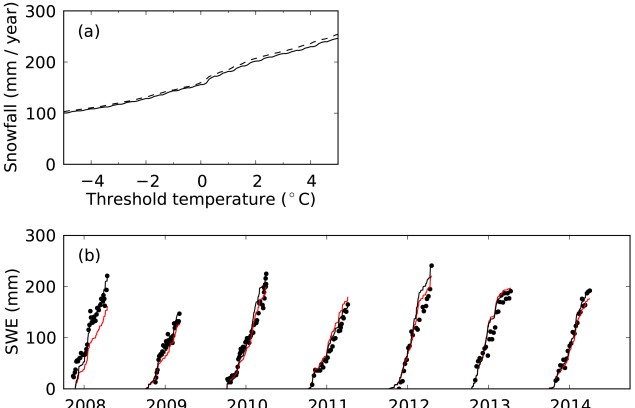

**Figure 3. (a)** Average annual snowfall derived from total precipitation with varying temperature (solid line) or wet-bulb temperature (dashed line) thresholds. **(b)** SWE on the ground from manual observations up to the maximum each winter (dots), cumulated snowfall (red lines) and snowfall scaled to the annual maxima (black lines).

**GID**

doi:10.5194/gi-2015-35

**A 7-year dataset for driving and evaluating snow models**

R. Essery et al.

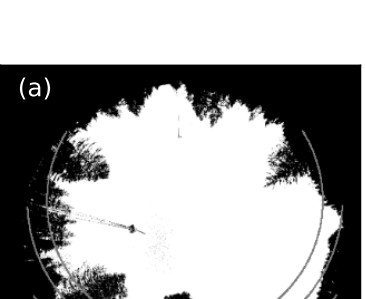

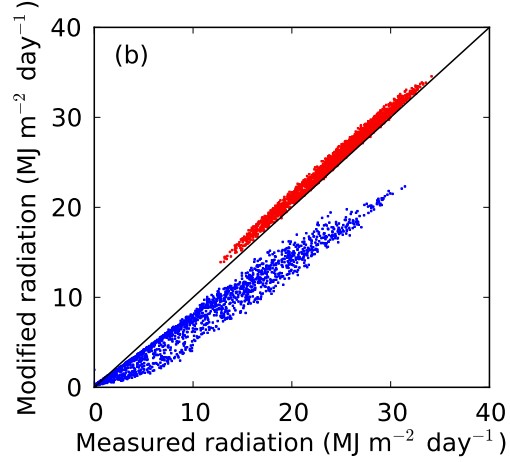

**Figure 4. (a)** A hemispherical photograph taken close to the IOA snow depth sensor in August 2011, showing the track of the sun (grey lines) on the first days of February, March, April and May. **(b)** Measured above-canopy and modified below-canopy daily solar (blue points) and longwave (red points) radiation.



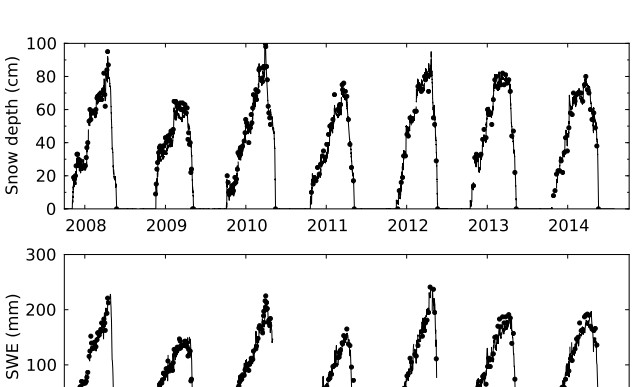


**Figure 5.** Measured snow depth and SWE from manual measurements (dots) and automatic instruments (lines). Daily averages of the automatic SWE measurements are used to reduce noise.

# GID

doi:10.5194/gi-2015-35

**A 7-year dataset for driving and evaluating snow models**

R. Essery et al.

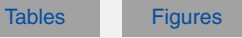
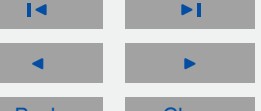
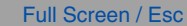

## GID

doi:10.5194/gi-2015-35

**A 7-year dataset for driving and evaluating snow models**

R. Essery et al.

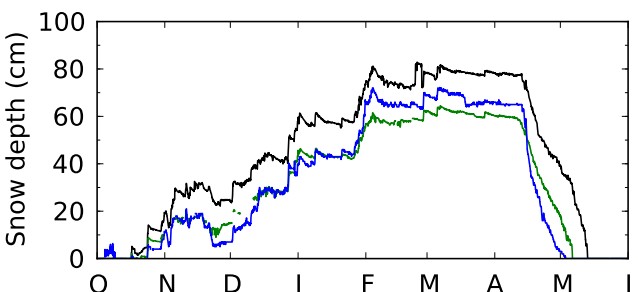

**Figure 6.** Snow depths measured in the IOA (black line), in the forest (green line) and on the wetland (blue line) for the winter of 2012–2013.

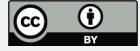

Discussion Paper | Discussion Paper | Discussion Paper | Discussion Paper | Discussion Paper |

# GID

doi:10.5194/gi-2015-35

**A 7-year dataset for driving and evaluating snow models**

R. Essery et al.

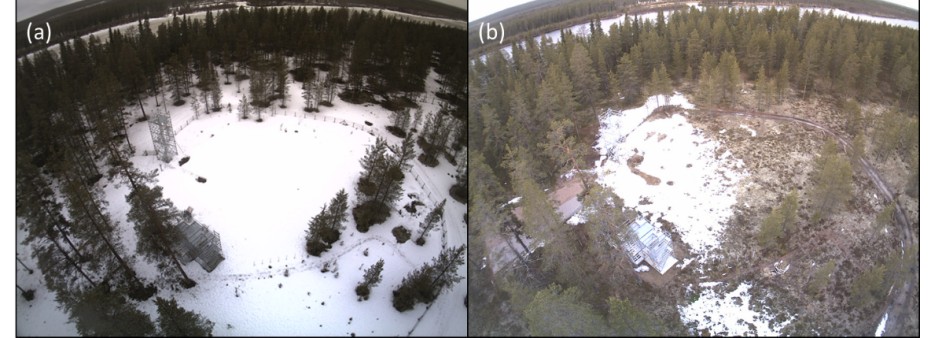

**Figure 7.** Snow melting in the IOA at noon on **(a)** 1 May and **(b)** 13 May 2013.

GID

doi:10.5194/gi-2015-35

A 7-year dataset for driving and evaluating snow models

R. Essery et al.

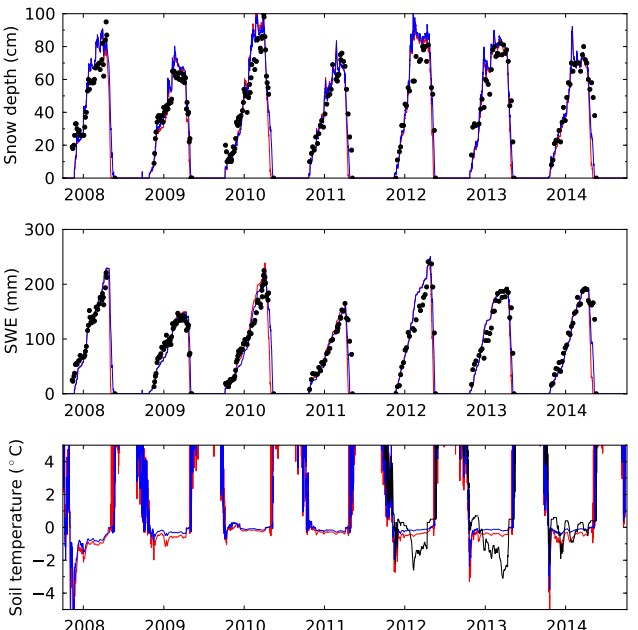

**Figure 8.** Crocus simulations with the above-canopy (red lines) and below-canopy (blue lines) driving datasets, compared with measurements (black dots and lines) of snow depth, SWE and soil temperature. For clarity, only manual measurements of snow depth and SWE are shown.

**GID**

doi:10.5194/gi-2015-35

A 7-year dataset for driving and evaluating snow models

R. Essery et al.

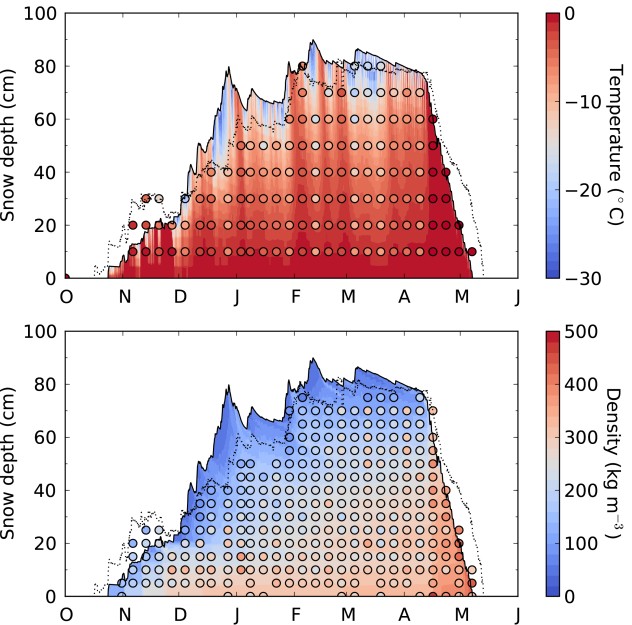

**Figure 9.** Profiles of snow temperature and density from Crocus simulations (background colours) and snow pit measurements (coloured dots) for the winter of 2012–2013. Dotted lines show the measured snow depth.

# GID

doi:10.5194/gi-2015-35

**A 7-year dataset for driving and evaluating snow models**

R. Essery et al.

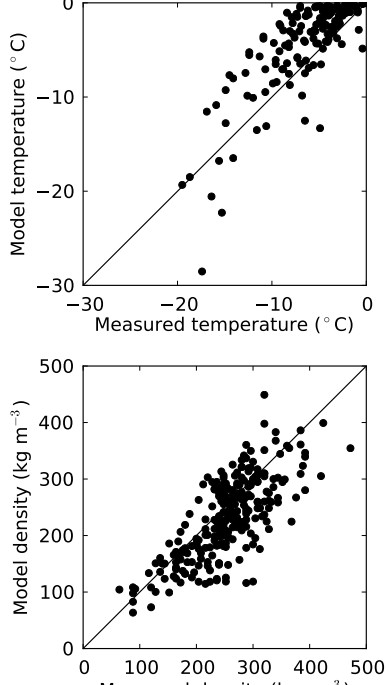

**Figure 10.** Scatter plots of Crocus simulations and manual snow pit measurements of snow temperature and density for the winter of 2012–2013.