# Peer review of "A 7-year dataset for driving and evaluating snow models at an arctic site (Sodankylä, Finland)"

_Geoscientific Instrumentation, Methods and Data Systems, 2015_

## Referee Comment (RC1) · C. Fierz (Referee) · 22 Feb 2016

GENERAL COMMENTS

This nicely written paper presents in a concise way data sets that can be used to drive and evaluate snow-cover models in an Arctic environment. In addition, above-canopy radiation and wind measurements have been modified in a separate set to also account for below-canopy conditions. This makes the sets particularly valuable but also shows that carefully evaluated procedures based on long-term records are required to do so, which may not be fully the case regarding wind speeds. Another example is the scaling applied to snowfall that would not be as convincing if applied to one or two winters only. In my view, however, the weakest point is the comparison of measurements with model simulations. While the scatter plots of Figure 10 may give the impression of a fair

correlation between those two, a look at the supplementary material shows that there is still work to do. But one has to take in account that quantitative comparisons of simulations with measurements are still quite difficult to perform.

I therefore recommend accepting the paper after the authors addressed the minor issues below.

SPECIFIC COMMENTS

p. 1, line 11: Is "bulk" needed here? Snow depth and snow water equivalent are intrinsically related to the snow-cover as a whole anyway.

p. 1, line 25: I wonder whether it would be better to cite the data set directly? WSL Institute for Snow and Avalanche Research SLF: Meteorological and snowpack measurements from Weissfluhjoch, Davos, Switzerland, Dataset, doi:10.16904/1, 2015 [first cited in: Wever, N., Schmid, L., Heilig, A., Eisen, O., Fierz, C. and Lehning, M.: Verification of the multi-layer SNOWPACK model with different water transport schemes, The Cryosphere, 9(6), 2271–2293, doi:10.5194/tc-9-2271-2015, 2015.]

p. 4, line 14: Equally important would be to know whether the instruments are ventilated.

p. 4, line 20: How far are AWS and radiometer tower apart? It may also be nice to mark the tower in Figure 1a.

p. 5, line 25: Do you apply any undercatch correction to precipitation data? From Table 1 it also appears the height of the precipitation gauge is surprisingly low (1 m) compared to WMO standard (1.5 m). Is there a reason for it?

p. 7, line 3: The increase in longwave radiation does not seem to be that substantial given the canopy. Can you comment on that?

p. 7, lines 11-12 : But wind adjustment will influence turbulent fluxes, will it not? Is this negligible?

p. 7, line 17: Are turbulent fluxes at FMI-ARC never directed towards the snow cover?

p. 8, lines 20-21: 'supported on a stick' It would be nice to know how the thermistors are mounted though. Maybe you could cite another publication, as this paper is not the place to describe that design in detail. A photograph showing the depression in the snow cover could suffice too?

p. 8 line 25: The air will affect pit temperatures at all heights. Why is the base that different?

p. 8, line 26: What volume? What cutter type is used?

Figure S1: Change 'density' to 'temperature' in caption

Figures S1  S2: I'd label the ordinate (y-axis) as 'Height' rather than 'Snow depth', which is total snow height.

Figures S1  S2: Reverse order in supplement

***** NOTE: The name of Marie Dumont is misspelled in the author list appearing on the web

---

## Referee Comment (RC2) · M. Raleigh (Referee) · 24 Feb 2016

GENERAL COMMENTS

The authors present a seven-year dataset of observed meteorology and snowpack conditions at a site in Finland for the purposes of snow model evaluation. Many recent published datasets at snow sites have been at mid-latitude mountain sites, so this high-latitude dataset is a novel contribution for testing and developing snow models in arctic environments. The authors provide sufficient and succinct descriptions of the site, measurements, and data processing/preparation, and demonstrate the usefulness of the data for running and evaluating the one-dimensional, multi-layer Crocus snow model. The authors include two meteorological datasets, one with wind and radiation measured above the canopy and the other with wind/radiation measurements adjusted

for the sub-canopy. These two datasets make the site data more broadly applicable to a variety of physical snow models (as some do not represent forest canopy). Given the novelty, length, and apparent quality of the data and the excellent presentation, I recommend this paper for publication pending minor revisions (see below).

SPECIFIC COMMENTS

- The driving and evaluation datasets are posted on the website of the Finnish Meteorological Institute (litdb.fmi.fi). What assurances are in place for long-term hosting of these data? Would it be prudent to have the data in a repository such as Pangea (e.g., Morin et al., 2012) or to have the datasets included in the supplemental material associated with the paper (e.g., Landry et al., 2014)?

- It would be useful for future analyses of the forcing data if the meteorological datasets included flags indicating whether the data for each variable is original or filled at each time step (see Landry et al., 2014). Can you please include these flags?

- The posted datasets include most of the evaluation data (SWE, snow depth, temperature profiles) described but not all those found in Table 2. Will the other evaluation data (e.g., snowpit measurements of density and temperature) also be made available?

- In the abstract, please state what years are included in the datasets.

- In the modified dataset ('mod_2007-10-01_2014-09-30.csv'), there are four time steps on 11 April 2010 with negative shortwave values. Please correct these data.

- In comparing the above and below canopy shortwave radiation, I found 1244 time steps where below canopy shortwave is greater than above canopy shortwave, which does not seem to make sense to me. Of those 1244 steps, 828 are when above canopy shortwave = 0, 412 are when above canopy shortwave is between 0 and 10 W/m2, and 4 are when above canopy shortwave is above 10 W/m2. I do not expect that below canopy shortwave can be greater than above canopy shortwave. I assume that most of these inconsistencies are due to processing (e.g., equation 1) and recommend

applying limits on the adjustments to rectify this situation.

- Please state the regularity with which the site is maintained/visited by FMI staff/technicians to check the integrity of instruments and check for measurement errors (e.g., snow on radiometer domes).

- Are the pyranometer and pyrgeometer heated/ventilated? Please clarify in the text. If not, do you have a sense of how often these radiometers were covered in snow or ice and how this may have influenced the quality and completeness of the radiation data? If this is a concern, there are methods for identifying periods of snow cover on the radiometer domes (e.g., Lapo et al., 2015).

- Is the temperature sensor naturally ventilated or mechanically ventilated? Please clarify in the text.

- Is the anemometer heated? Based on specifications, I think it is heated, but it would be good to state this in the text to allay concerns about freezing of the anemometer.

TECHNICAL COMMENTS/CORRECTIONS

- Pg. 2, Lines 22-25: Please consider including a reference to the Wayand et al. (2015) snow and meteorology dataset in the Washington Cascades as another recent example.

- Pg. 3, Line 2: The FMI-ARC site is not only unique for being at a high latitude, but also at a low elevation (with snowfall temperatures close to 0 C, Pg. 6) and having a more shallow snowpack than the other snow sites with published datasets. It might be worth highlighting these distinct features here.

- Pg. 4, Line 2: A minor suggestion: it would be interesting to briefly summarize the data going back to 1908 and compare to the equivalent data summaries over the period represented in this dataset (Oct 2007 – Sept 2014). This would provide context, e.g., which years in this dataset had low/average/high precipitation in terms of the longer data record.

- Pg. 5, Line 12: The supplement lists the ERA-Interim bias as -5.1 W/m2 (relative to observations). Please clarify whether any bias correction was applied to the ERA-Interim data and why/why not.

- Pg. 5, Line 16: Please replace "Raleigh et al., 2015a" with "Raleigh et al., 2016" and update the citations in the references.

- Pg. 6, Lines 2-10: Did this scale factor vary annually? Can you please specify the scale factor(s) here?

- Pg. 6, Lines 24-25: There are potential problems with estimating canopy temperature from air temperature in some situations (see Webster et al., 2016). Please comment on this briefly.

- Pg. 9, lines 26-28: Can you clarify whether the soils are freezing in these cases (as suggested by the observed soil temperatures)? Is anything known about the soil moisture?

- Table 1: Air temperature and precipitation are both measured at 1 m. The maximum snow depth in the dataset is 1.02 m (late March 2010) and in five winters the maximum depth exceeds 0.80 m. Given spatial variability in snow depth, it begs the question whether the temperature sensor and precipitation gauge were ever buried in snow during the observational period. Please comment.

- Table 1: Does the HMP35D measure temperature in addition to humidity? If so, was this used to double check and fill missing data from the PTB201A?

- Figure 1: It would be helpful to include another panel with a map of Finland and a marker showing the site location in the country.

- Figure 3b: The two lines are quite thin and difficult to distinguish to my eye. Can you please take some measures to help me discriminate them better?

- Figure 8: What depth are the soil temperature simulations / observations in this figure? Please clarify in the caption and/or figure.

- Supplement: The caption of the third figure (multiple pages) with temperature profiles has two errors. First, the figure is called "Figure S1" when it should be "Figure S3". Second, the caption should state "Profiles of snow temperature" instead of "Profiles of snow density".

CITATIONS

Lapo, K. E., L. M. Hinkelman, C. C. Landry, A. K. Massmann, and J. D. Lundquist (2015), A simple algorithm for identifying periods of snow accumulation on a radiometer, Water Resour. Res., 51(9), 7820–7828, doi:10.1002/2015WR017590.

Raleigh, M. S., B. Livneh, K. Lapo, and J. D. Lundquist (2016), How does availability of meteorological forcing data impact physically-based snowpack simulations?, J. Hydrometeorol., 17(1), 99–120, doi:10.1175/JHM-D-14-0235.1.

Wayand, N. E., A. Massmann, C. Butler, E. Keenan, J. Stimberis, and J. D. Lundquist (2015), A meteorological and snow observational data set from Snoqualmie Pass (921 m), Washington Cascades, USA, Water Resour. Res., 51(12), 10092–10103, doi:10.1002/2015WR017773.

Webster, C., N. Rutter, F. Zahner, and T. Jonas (2016), Modeling sub-canopy incoming longwave radiation to seasonal snow using air and tree trunk temperatures, J. Geophys. Res. Atmos., 121, doi:10.1002/2015JD024099.

---

## Author Comment (AC1) · 8 Apr 2016

We thank Charles Fierz and Mark Raleigh for their comments, which are reproduced in italic font below with responses in roman font.

**Referee comments from Charles Fierz**

*the weakest point is the comparison of measurements with model simulations. While the scatter plots of Figure 10 may give the impression of a fair correlation between those two, a look at the supplementary material shows that there is still work to do. But one has to take in account that quantitative comparisons of simulations with measurements are still quite difficult to perform.*
Indeed, there is much more work to do. This paper is about providing data necessary for that work, rather than directly about improving snow simulations. The simulations are included as an illustration of the data in use.

*p. 1, line 11: Is "bulk" needed here? Snow depth and snow water equivalent are intrinsically related to the snow-cover as a whole anyway.*
The word "bulk" is not needed in describing snow depth or snow water equivalent alone but is included here to emphasize the distinction between these measurements and vertically-resolved profile measurements.

*p. 1, line 25: I wonder whether it would be better to cite the data set directly? WSL Institute for Snow and Avalanche Research SLF: Meteorological and snowpack measurements from Weissfluhjoch, Davos, Switzerland, Dataset, doi:10.16904/1, 2015*
Done

*p. 4, line 14: Equally important would be to know whether the instruments are ventilated.*
The radiometers are mechanically ventilated but the temperature and humidity sensors are not.

*p. 4, line 20: How far are AWS and radiometer tower apart? It may also be nice to mark the tower in Figure 1a.*
They are 30 m apart, and are now marked on Figure 1b.

*p. 5, line 25: Do you apply any undercatch correction to precipitation data? From Table 1 it also appears the height of the precipitation gauge is surprisingly low (1 m) compared to WMO standard (1.5 m). Is there a reason for it?*
In fact, the precipitation data were taken from an optical gauge mounted at 2 m height to remain clear of the snow; this information has been corrected in Table 1 and added in the text. No wind correction is applied to the precipitation data, but wind speeds are generally low and the adjustment to match snow accumulation on the ground will correct any undercatch.

*p. 7, line 3: The increase in longwave radiation does not seem to be that substantial given the canopy. Can you comment on that?*
Snow measurements are made in a clearing but towards the shaded side where longwave radiation increases will be least. This is now discussed in more detail.

*p. 7, lines 11-12: But wind adjustment will influence turbulent fluxes, will it not? Is this negligible?*
The wind adjustment only makes a small difference in simulations with Crocus, but other models may be more sensitive. It is precisely because of the potential influence on turbulent fluxes that the wind adjustment is applied.

*p. 7, line 17: Are turbulent fluxes at FMI-ARC never directed towards the snow cover?*
This has been rephrased as "sensible and latent heat exchanges with the atmosphere" to avoid the impression that they are only in one direction.

*p. 8, lines 20-21: 'supported on a stick' It would be nice to know how the thermistors are mounted though. Maybe you could cite another publication, as this paper is not the place to describe that design in detail. A photograph showing the depression in the snow cover could suffice too?*
A link is now given to the page on the Sodankylä data archive for this system, which has photographs.

*p. 8 line 25: The air will affect pit temperatures at all heights. Why is the base that different?*
This has been rephrased as "digging a snow pit brings air into contact with snow beneath the surface".

*p. 8, line 26: What volume? What cutter type is used?*
Information that 250 and 500 cm$^3$ cutters are used and a reference (Leppänen et al. 2015, this volume) with more details have been added to the text.

*Figure S1: Change 'density' to 'temperature' in caption*
Done (and number corrected to S3)

*Figures S1 S2: I'd label the ordinate (y-axis) as 'Height' rather than 'Snow depth', which is total snow height.*
Done

*Figures S1 S2: Reverse order in supplement*
Corrected

***** NOTE: The name of Marie Dumont is misspelled in the author list appearing on the web

**Referee comments from Mark Raleigh**

*- The driving and evaluation datasets are posted on the website of the Finnish Meteorological Institute (litdb.fmi.fi). What assurances are in place for long-term hosting of these data?*
Under the open data policy of the Finnish Ministry of Transport and Communications, FMI is committed to the long-term upkeep and public distribution of its datasets. This assurance has been added to the text.

*- It would be useful for future analyses of the forcing data if the meteorological datasets included flags indicating whether the data for each variable is original or filled at each time step (see Landry et al., 2014). Can you please include these flags?*
A flag has been added to the forcing data files to record which variables had to be filled at each timestep, and this is now stated in the text.

*- The posted datasets include most of the evaluation data (SWE, snow depth, temperature profiles) described but not all those found in Table 2. Will the other evaluation data (e.g., snowpit measurements of density and temperature) also be made available?*
All of the datasets are listed on http://litdb.fmi.fi/index.php and are either directly available or available on request. It is now pointed out that contact information for each dataset is given in the database.

*- In the abstract, please state what years are included in the datasets.*
Done

*- In the modified dataset ('mod_2007-10-01_2014-09-30.csv'), there are four time steps on 11 April 2010 with negative shortwave values. Please correct these data.*
Corrected

*- In comparing the above and below canopy shortwave radiation, I found 1244 time steps where below canopy shortwave is greater than above canopy shortwave, which does not seem to make sense to me.*
We are very grateful to Dr Raleigh for spotting these data processing errors, which have been corrected. Fortunately, the corrections have had minimal impacts on snow simulations.

*- Please state the regularity with which the site is maintained/visited by FMI staff/technicians to check the integrity of instruments and check for measurement errors (e.g., snow on radiometer domes).*
Now stated in the text – the site is staffed 5 days a week and instruments are checked after every snowfall, when automatic error checking identifies a problem or at least 3 times a week.

*- Are the pyranometer and pyrgeometer heated/ventilated? Please clarify in the text.*
Clarified in the text – they are ventilated. Comparison of incoming and outgoing shortwave radiation suggests that unnoticed covering of the radiometers by snow is very rare.

*- Is the temperature sensor naturally ventilated or mechanically ventilated? Please clarify in the text.*
The text now states that the temperature sensor ventilation is natural.

*- Is the anemometer heated? Based on specifications, I think it is heated, but it would be good to state this in the text to allay concerns about freezing of the anemometer.*
Yes – now stated in the text

*- Pg. 2, Lines 22-25: Please consider including a reference to the Wayand et al. (2015) snow and meteorology dataset in the Washington Cascades as another recent example.*
Done

*- Pg. 3, Line 2: The FMI-ARC site is not only unique for being at a high latitude, but also at a low elevation (with snowfall temperatures close to 0 C, Pg. 6) and having a more shallow snowpack than the other snow sites with published datasets. It might be worth highlighting these distinct features.*
There are actually several mid-latitude continental sites of low relief with published snow data; the distinction of the FMI-ARC site is its high latitude.

*- Pg. 4, Line 2: A minor suggestion: it would be interesting to briefly summarize the data going back to 1908 and compare to the equivalent data summaries over the period represented in this dataset (Oct 2007 – Sept 2014). This would provide context, e.g., which years in this dataset had low/average/high precipitation in terms of the longer data record.*
A reference (Kivi et al. 1999) which includes a figure showing temperature changes since 1908 has been added. Statistics of annual maximum snow depth over 50 years have been added to provide context.

*- Pg. 5, Line 12: The supplement lists the ERA-Interim bias as -5.1 W/m2 (relative to observations). Please clarify whether any bias correction was applied to the ERAInterim data and why/why not.*
It is now explicitly stated that the bias has been removed.

*- Pg. 5, Line 16: Please replace "Raleigh et al., 2015a" with "Raleigh et al., 2016" and update the citations in the references.*
Done

*- Pg. 6, Lines 2-10: Did this scale factor vary annually? Can you please specify the scale factor(s) here?*
Annual scaling factors are now specified in a table.

*- Pg. 6, Lines 24-25: There are potential problems with estimating canopy temperature from air temperature in some situations (see Webster et al., 2016). Please comment on this briefly.*
This is now commented on as being more of a problem towards the sun-lit side of the clearing, and a reference on the issue (Pomeroy et al. 2009) has been added.

*- Pg. 9, lines 26-28: Can you clarify whether the soils are freezing in these cases (as suggested by the observed soil temperatures)? Is anything known about the soil moisture?*
A reference (Rautiainen et al. 2014) has been added showing that soil freezing can exceed 2 m in depth. It is now mentioned that soil moisture is measured but not used in this paper.

*- Table 1: Air temperature and precipitation are both measured at 1 m. The maximum snow depth in the dataset is 1.02 m (late March 2010) and in five winters the maximum depth exceeds 0.80 m. Given spatial variability in snow depth, it begs the question whether the temperature sensor and precipitation gauge were ever buried in snow during the observational period. Please comment.*
The air temperature and precipitation sensors are both mounted at 2 m height above the ground and have never been buried in snow.

*- Table 1: Does the HMP35D measure temperature in addition to humidity? If so, was this used to double check and fill missing data from the PTB201A?*
The HMP35D temperature output is not recorded.

*- Figure 1: It would be helpful to include another panel with a map of Finland and a marker showing the site location in the country.*
Done

*- Figure 3b: The two lines are quite thin and difficult to distinguish to my eye. Can you please take some measures to help me discriminate them better?*
The essential information in this figure is the mismatch between cumulated snowfall measurements and maximum SWE measured on the ground, so the red lines have been replaced by dots that show the cumulated snowfall up to the time of maximum SWE. There are no longer two lines that need to be distinguished.

*- Figure 8: What depth are the soil temperature simulations / observations in this figure? Please clarify in the caption and/or figure.*
Added to the caption

*- Supplement: The caption of the third figure (multiple pages) with temperature profiles has two errors. First, the figure is called "Figure S1" when it should be "Figure S3". Second, the caption should state "Profiles of snow temperature" instead of "Profiles of snow density".*
Corrected

Manuscript prepared for Geosci. Instrum. Method. Data Syst.
with version 2015/09/17 7.94 Copernicus papers of the LATEX class copernicus.cls.
Date: 8 April 2016

[revised manuscript text omitted]

**2  Site**

FMI-ARC (67.368°N, 26.633°E, 179 m above sea level, Figure 1a) is collocated with the Sodankylä Geophysical Observatory beside the Kitinen river, 90 km north of the Arctic Circle and 7 km south-east of the town of Sodankylä in northern Finland. Snow typically lies from October until May; in daily records between 1951 and 2000, the annual maximum snow depth had a median of 83 cm, an interquartile range of 21 cm and a range from 62 cm (1954) to 119 cm (2000). Soil frost depths can reach over 2 m (Rautiainen et al., 2014) and air temperatures can fall below -30°C in winter, but the sun only remains entirely below the horizon for a few days in December. Continuous meteorological measurements have been made at or near this site since 1908 (Kivi et al., 1999). Current instrumentation includes an automatic weather station and an upper air sounding station (World Meteorological Organization index number 02836) which transmit data on the Global Telecommunication System for use by numerical weather prediction centres. In addition to regular measurement programmes, the Sodankylä area has been used in many remote sensing missions and field campaigns, including the Nordic Snow Radar Experiment (NoSREx), the Snow Reflectance Transition Experiment (SnoRTEx) and the Solid Precipitation Intercomparison Experiment (SPICE).

Figure 1b is an aerial orthophotograph of the site. The area around FMI-ARC is level and forested, predominantly with pine trees about 15 m tall, but many measurements are made in clearings or in a large wetland area to the east of the site. Driving data for this paper are taken from an automatic weather station (Figure 1c) and a radiometer tower (Figure 1d) 30 m apart, with instruments that are calibrated annually. Evaluation data are taken from an Intensive Observation Area (Figure 1e) that was established 590 m to the south of the weather station for NoSREx (Lemmetyinen et al., 2015). A list of many other observations not discussed in this paper and contact information can be found at http://litdb.fmi.fi/index.php.

**3  Driving data and gap filling**

All of the meteorological variables necessary for model driving are measured by the AWS and the radiometer tower at FMI-ARC with the instruments and at the heights listed in Table 1; note that radiation and wind measurements are made at heights above the forest canopy. The radiometers are ventilated and the anemometer is heated to reduce problems with freezing or snow accumulation, and instruments are cleaned after every snowfall or at least three times a week. Temperature and humidity sensors are naturally ventilated inside a Stevenson screen. Precipitation is measured using an optical sensor and two weighing gauges which give similar total amounts; data from the optical sensor are used here. There is no nearby wind speed measurement that could be used for gauge correction, but wind speeds are generally low and measured snowfall has been adjusted to match snow accumulation on the ground as described below. FMI-ARC is staffed five days a week, and automatic error checking can identify instrument problems immediately. For the 7-year period collated here, less than 1% of hourly data (visible as red points in Figure 2) are missing for any variable with the exception of longwave radiation; the longest period of missing data is a 52-day gap in the longwave radiation measurements from 10 September to 31 October 2011 because of a faulty power supply. The archived driving data files include a flag that records which data were missing and had to be filled for each hour.

Measurements from the AWS and the radiometer tower are used for driving data whenever they are available, but gaps have to be filled to form a complete driving dataset. Gaps of four hours or shorter are filled by linear interpolation. For shortwave radiation, air temperature, humidity and wind speed, longer gaps are filled with data from nearby instruments. No alternative longwave radiometer was operating at FMI-ARC for the full period, so longwave gaps are filled using ERA-Interim reanalyses (Dee et al., 2011). Longwave radiation fluxes in ERA-Interim are produced by short-range forecasts that can be expected to be accurate if the analysed vertical profiles of temperature and humidity in the atmosphere are accurate, although errors may be larger in cloudy conditions (Kangas et al., 2015). Data from both the surface synoptic station and the upper air station at Sodankylä are available for assimilation in reanalyses, and ERA-Interim compares well with the in situ measurements; the

longwave radiation measurements and forecasts have a correlation coefficient of 0.88 and a root mean square difference of 26.2 W m$^{-2}$ after removal of a 5.1 W m$^{-2}$ bias for periods when both are available (a scatter plot is included as supplementary material). Direct measurements of longwave radiation are rarely available for cold regions and snow models are known to be sensitive to longwave driving data (Raleigh et al., 2016); having near-continuous longwave measurements is therefore a distinct advantage of the FMI-ARC site.

[revised manuscript text omitted]

Raleigh, M. S., B. Livneh, K. Lapo and J. D. Lundquist: How does availability of meteorological forcing data
impact physically-based snowpack simulations? J. Hydrometeorol., 17(1), 99-120, doi:10.1175/JHM-D-14-
0235.1, 2016.

Rautiainen, K., J. Lemmetyinen, M. Schwank, A. Kontu, C. B. Ménard, C. Mätzler, M. Drusch, A. Wiesmann,
J. Ikonen, J. Pulliainen: Detection of soil freezing from L-band passive microwave observations, Remote
Sens. Environ., 147, 206-218, doi:10.1016/j.rse.2014.03.007, 2014.

Reba, M. L., D. Marks, M. Seyfried, A. Winstral, M. Kumar and G. Flerchinger: A long-term data
set for hydrologic modeling in a snow-dominated mountain catchment, Water Resour. Res., 47,
doi:10.1029/2010WR010030, 2011.

Reid, T.D., R. L. H. Essery, N. Rutter and M. King: Data-driven modelling of shortwave radiation transfer to
snow through boreal birch and conifer canopies, Hydrol. Processes, 28, 2987-3007, doi:10.1002/hyp.9849,
2014.

Sihvola, A., and M. Tiuri: Snow Fork for field determination of the density and wetness profiles of a snow pack,
IEEE Trans. Geosci. Remote Sens., 5, 717-721, 1986

Sims, E. M., and G. Liu: A Parameterization of the probability of snow-rain transition, J. Hydrometeorol., 16,
1466-1477, doi:http://dx.doi.org/10.1175/JHM-D-14-0211.1, 2015.

Slater, A. G., and 32 others: The representation of snow in land surface schemes: results from PILPS 2(d), J.
Hydrometeorol., 2, 7-25, 2001.

Vionnet, V., E. Brun, S. Morin, A. Boone, S. Faroux, P. Le Moigne, E. Martin and J.-M. Willemet: The de-
tailed snowpack scheme Crocus and its implementation in SURFEX v7.2, Geosci. Model Dev., 5, 773-791,
doi:10.5194/gmd-5-773-2012, 2012.

Wayand, N. E., A. Massmann, C. Butler, E. Keenan, J. Stimberis and J. D. Lundquist: A meteorological and
snow observational data set from Snoqualmie Pass (921 m), Washington Cascades, USA, Water Resour. Res.,
51, 10092-10103, doi:10.1002/2015WR017773, 2015.

WSL Institute for Snow and Avalanche Research SLF: Meteorological and snowpack measurements from
Weissfluhjoch, Davos, Switzerland, Dataset, doi:10.16904/1, 2015.

**Table 1.** Instruments and missing data for meteorological driving variables between 1 October 2007 and 30 September 2014

| Variable | Instrument | Height | Missing data |
|---|---|---|---|
| Precipitation | Vaisala FD12P | 2 m | 0.67% |
| Air pressure | Vaisala PTB201A | 1 m | 0.11% |
| Air temperature | Pentronic PT100 | 2 m | 0.11% |
| Relative humidity | Vaisala HMP35D | 2 m | 0.19% |
| Global solar radiation | Kipp & Zonen CM11 | 14 m | 0.61% |
| Diffuse solar radiation | Kipp & Zonen CM11 with shading ball | 14 m | 0.55% |
| Longwave radiation | Kipp & Zonen CG4 | 14 m | 8.65% |
| Wind speed | Vaisala WAA25 | 22 m | 0.12% |

**Table 2.** Scaling factors required to match measured snowfall to measured snow accumulation

| Winter | Factor |
|---|---|
| 2007-2008 | 1.373 |
| 2008-2009 | 1.165 |
| 2009-2010 | 1.131 |
| 2010-2011 | 0.922 |
| 2011-2012 | 1.093 |
| 2012-2013 | 0.971 |
| 2013-2014 | 1.092 |

**Table 3.** Evaluation data from the IOA

| Variable | Instrument |
|---|---|
| Snow depth | Campbell Scientific SR50
Manual sampling |
| Snow water equivalent | Astrock Gamma Water Instrument
Manual sampling |
| Snow density profiles | Toikka Snow Fork sampling at 10 cm height increments from 9/10/2009
Manual sampling at 5 cm height increments from 7/12/2009 |
| Snow temperature profiles | Campbell Scientific 107-L at 10 cm height increments from 6/9/2011
Manual sampling at 10 cm height increments |
| Soil temperature profiles | Decagon Devices 5TE at 5, 10, 20, 40 and 80 cm depths from 6/9/2011 |

[Figure]

**Figure 1.** (a) The location of FMI-ARC (dot), 90 km north of the Arctic Cirle (dashed line) in Finland. (b) Orthophotograph of the FMI-ARC site, showing the locations of the automatic weather station (AWS), the radiometer tower (rad) and the Intensive Observation Area (IOA). (c) The automatic weather station, with the radiometer tower in the background. (d) The radiometer tower. (e) The IOA, showing the locations of the ultrasonic depth gauge (UDG), the snow temperature profile and the snow pit for manual measurements.

[Figure]

**Figure 2.** Hourly timeseries of shortwave radiation, longwave radiation, air temperature, relative humidity, wind speed and pressure. Longwave radiation data points in red are from reanalyses.

[Figure]

**Figure 3.** (a) Average annual snowfall derived from total precipitation with varying temperature (solid line) or wet-bulb temperature (dashed line) thresholds. (b) SWE on the ground from manual observations up to the maximum each winter (black dots), cumulated snowfall up to the date of maximum SWE (white dots) and snowfall scaled to the annual maxima (black lines).

[Figure]

**Figure 4.** (a) A hemispherical photograph taken close to the IOA snow depth sensor in August 2011, showing the track of the sun (grey lines) on the first days of February, March, April and May. (b) Measured above-canopy and modified below-canopy daily solar (blue points) and longwave (red points) radiation.

[Figure]

**Figure 5.** Measured snow depth and SWE from manual measurements (dots) and automatic instruments (lines). Daily averages of the automatic SWE measurements are used to reduce noise.

[Figure]

**Figure 6.** Snow depths measured in the IOA (black line), in the forest (green line) and on the wetland (blue line) for the winter of 2012-2013.

[Figure]

**Figure 7.** Snow melting in the IOA at noon on (a) 1 May and (b) 13 May 2013.

[Figure]

**Figure 8.** Crocus simulations with the above-canopy (red lines) and below-canopy (blue lines) driving datasets, compared with measurements (black dots and lines) of snow depth, SWE and soil temperature at 10 cm depth. For clarity, only manual measurements of snow depth and SWE are shown.

[Figure]

**Figure 9.** Profiles of snow temperature and density from Crocus simulations (background colours) and snow pit measurements (coloured dots) for the winter of 2012-2013. Dotted lines show the measured snow depth.

[Figure]

**Figure 10.** Scatter plots of Crocus simulations and manual snow pit measurements of snow temperature and density for the winter of 2012-2013.

**A 7-year dataset for driving and evaluating snow models at an arctic site (Sodankylä, Finland)**

Richard Essery, Anna Kontu, Juha Lemmetyinen, Marie Dumont and Cécile B Ménard

**Supplementary figures**

**Figure S1.** Scatter plot and statistics comparing in situ measurements and ERA-Interim forecasts of longwave radiation averaged over periods of 3 hours (blue dots) and a day (red dots).

[Figure]

| Period | Bias (W m$^{-2}$) | Root mean square error (W m$^{-2}$) | Correlation |
|---|---|---|---|
| 3 hours | -5.1 | 26.7 | 0.88 |
| 24 hours | -5.1 | 16.9 | 0.94 |

**Figure S2.** Profiles of snow density from manual measurements (black dots), dielectric measurements (white dots) and Crocus simulations (lines).

[Figure]

[Figure]

[Figure]

**Figure S2.** Continued

[Figure]

[Figure]

**Figure S2.** Continued

[Figure]

[Figure]

**Figure S3.** Profiles of snow temperature from manual measurements (black dots), thermistors (white dots) and Crocus simulations (lines).

[Figure]

[Figure]

**Figure S3.** Continued

[Figure]

**Figure S3.** Continued

[Figure]

**Figure S3.** Continued

[Figure]

**Figure S3.** Continued

[Figure]

**Figure S3.** Continued

[Figure]